# High VEGFA Expression Is Associated with Improved Progression-Free Survival after Bevacizumab Treatment in Recurrent Glioblastoma

**DOI:** 10.3390/cancers15082196

**Published:** 2023-04-07

**Authors:** Bárbara Alves, Joana Peixoto, Sofia Macedo, Jorge Pinheiro, Bruno Carvalho, Paula Soares, Jorge Lima, Raquel T. Lima

**Affiliations:** 1i3S—Instituto de Investigação e Inovação em Saúde, 4200 Porto, Portugal; barbaracvalves@gmail.com (B.A.); jpeixoto@ipatimup.pt (J.P.); amacedo@ipatimup.pt (S.M.); psoares@ipatimup.pt (P.S.);; 2Cancer Signalling & Metabolism Group, IPATIMUP, Institute of Molecular Pathology and Immunology of the University of Porto, 4200 Porto, Portugal; 3School of Allied Health Sciences, Polytechnic Institute of Porto, 4200 Porto, Portugal; 4Department of Pathology, Centro Hospitalar Universitário S. João, 4200 Porto, Portugal; 5Department of Neurosurgery, Centro Hospitalar Universitário S. João, 4200 Porto, Portugal; bmfcarvalho@med.up.pt; 6FMUP—Faculty of Medicine of the University of Porto, 4200 Porto, Portugal; 7Department of Pathology, FMUP—Faculty of Medicine of the University of Porto, 4200 Porto, Portugal

**Keywords:** glioblastoma, secretome, biomarkers, immunohistochemistry, VEGFA, YKL40, MMP-2, MMP-9

## Abstract

**Simple Summary:**

Glioblastoma (GB) is a deadly tumor that demands for relevant biomarkers, particularly regarding patients’ response to treatment. MMP-2, MMP-9, VEGFA, and YKL40 are important molecules, given their implication in the infiltrative and angiogenic phenotype of GBs. The purpose of this study was to assess the relationship between the expression of MMP-2, MMP-9, VEGFA, and YKL40 in GB tissues and the patients’ response to temozolomide (first-line treatment) or bevacizumab (second-line treatment). Our results showed that increased VEGFA is significantly associated with an improved response to bevacizumab, while having no correlation with the response to temozolomide. Additionally, YKL40 expression may also be important regarding information about the extent of antiangiogenic treatment in GB patients.

**Abstract:**

Glioblastoma (GB) is one of the deadliest human cancers. Many GB patients do not respond to treatment, and inevitably die within a median of 15–18 months post-diagnosis, highlighting the need for reliable biomarkers to aid clinical management and treatment evaluation. The GB microenvironment holds tremendous potential as a source of biomarkers; several proteins such as MMP-2, MMP-9, YKL40, and VEGFA have been identified as being differentially expressed in GB patient samples. Still to date, none of these proteins have been translated into relevant clinical biomarkers. This study evaluated the expression of MMP-2, MMP-9, YKL40, and VEGFA in a series of GBs and their impact on patient outcome. High levels of VEGFA expression were significantly associated with improved progression-free survival after bevacizumab treatment, thus having potential as a tissue biomarker for predicting patients’ response to bevacizumab. Noteworthily, VEGFA expression was not associated with patient outcome after temozolomide treatment. To a lesser extent, YKL40 also provided significant information regarding the extent of bevacizumab treatment. This study highlights the importance of studying secretome-associated proteins as GB biomarkers and identifies VEGFA as a promising marker for predicting response to bevacizumab.

## 1. Introduction

Despite the current technological advances in imaging diagnosis and follow-up, as well as in surgical procedures, and in radio- and chemotherapeutic approaches, the prognosis of glioblastoma (GB) patients remains extremely poor [1], with a median overall survival (OS) of 15–18 months for patients receiving standard therapy [2,3], and 3–4 months for untreated patients [4]. Thus far, the Stupp protocol remains the standard regimen of multimodal treatment for GB. It involves maximal safe surgical resection of the tumor followed by radiotherapy and concomitant chemotherapy, typically with temozolomide (TMZ) [5]. TMZ, a chemotherapeutic drug which acts through DNA alkylation, causing DNA damage and cell death [6], has been shown to significantly improve GB patients’ OS and progression-free survival (PFS) compared to radiotherapy alone [6]. Still, patients recur, prompting the use of second-line agents such as bevacizumab antiangiogenic therapy, an antibody that targets the vascular endothelial growth factor (VEGFA) [7]. However, this alternative therapy is not effective in all patients, only improving PFS, but not OS [8,9].

Given the heterogeneity of GBs [10] and the limitations in diagnosis and patients’ follow-up, combined with the limited success of the current treatments [11,12], it is crucial to identify the patients who will benefit the most from a specific therapeutic approach. In this context, molecular biomarkers are a valuable tool to stratify patients in order to improve their outcomes, minimize the use of ineffective treatments, and to reduce the associated toxicity [13].

GB secretome holds tremendous potential not only by providing potential therapy targets, but also as a promising source of GB biomarkers. Several secretome proteins have been linked to the highly infiltrative and angiogenic phenotype of GBs [14,15], which are considered two key features of the disease [16]. These include MMP-2, MMP-9, VEGFA, and YKL40 [1,17,18,19,20]. The role of MMP-2 and MMP-9 in the degradation of ECM significantly contributes to tumor invasion and the aggressiveness of GBs [1], while the aberrant expression of VEGFA is suggested to drive angiogenesis and tumorigenesis [15,21]. YKL40 (also known as CHI3L1) has been described as having an important role in glioma cell proliferation [22]. Despite several studies evaluating the potential of these and other proteins as predictors of prognosis [23,24,25,26,27,28,29] and of response to treatment [25,30,31], we still lack robust results, and to date, none of them have been clinically implemented as biomarkers, particularly as treatment response predictors.

Given the relevance of these proteins in tumor development and their potential as clinically relevant biomarkers for patient management, the purpose of this study was to assess the prognostic potential of the secretome-associated proteins (MMP-2, MMP-9, VEGFA, and YKL40) in tissue samples from a cohort of patients with recurrent GB who had been subjected to the Stupp Protocol (first line), and to bevacizumab as second-line treatment upon recurrence under TMZ.

## 2. Materials and Methods

### 2.1. Patients and Samples

In this retrospective study, tumor tissue was used from a series of 63 patients from the Centro Hospitalar Universitário de São João (CHUSJ), collected between 2011 and 2017. Patients’ clinicopathological and molecular variables (including diagnosis and follow-up) and tumor samples were used under the approval of CHUSJ ethical committee (no. 99/14 and 201/21) and were retrieved from surgical pathology and patients’ records from the CHUSJ Neurosurgery department database.

The criteria for patients’ inclusion were: (i) histologically proven glioblastoma (according to the *WHO Classification of Tumors of the Central Nervous System* guidelines published in 2016 [32]); (ii) patient age ≥ 18 years; (iii) subjected either to tumor resection or biopsy. All patients were given radiotherapy and chemotherapy (75 mg/m^2^ of temozolomide) as first-line treatment, in accordance with the Stupp Protocol. As second-line treatment, bevacizumab-based therapy (10 mg/kg), either alone or in combination with irinotecan or lomustine, was given to all patients.

### 2.2. Immunohistochemistry Analysis in Human Tissue Samples

Immunohistochemistry (IHC) analysis was carried out in representative tumor tissue sections obtained from FFPE tissue blocks, cut into serial 4 µm-thick sections, and collected onto positively charged slides. Slides were heated at 60 °C for 15 min, deparaffinized and rehydrated through serial immersions in a set of xylol and ethanol (100%, 96% and 70%) solutions (all from Valente e Ribeiro, Lda, Sintra, Portugal). After permeabilization and rehydration, the following IHC procedures were carried out according to the antibodies utilized.

#### 2.2.1. MMP-2, MMP-9 and VEGFA Immunohistochemistry

Heat-induced antigen retrieval was carried out for 45 min in a steamer using Epitope Retrieval Solution pH 9 (Leica Microsystems, Wetzlar, Germany) for MMPs-2 and -9, and citrate buffer pH 6.0 (Leica Microsystems, Wetzlar, Germany) for VEGFA. After cooling down at room temperature, slides were incubated for 10 min with UltraVision Hydrogen Peroxide Block (Thermo Fisher Scientific, Waltham, MA, USA) for endogenous peroxidase blocking, and further incubated for 10 min with UltraVision Protein Block (Thermo Fisher Scientific, Waltham, MA, USA). For VEGFA, an additional 10 min blocking step with UltraBlock solution (Ventana Medical Systems, Oro Valley, AZ, USA) was included between the two previous steps. Tissue sections were then incubated overnight at 4 °C with the following antibodies: MMP-2 [ab866 (Abcam, Cambridge, UK); dilution 1:300], MMP-9 [ab76003 (Abcam, Cambridge, UK); dilution 1:1000], and VEGFA [HPA069116 (Sigma-Aldrich, St. Louis, MO, USA); dilution 1:400]. Primary antibodies were omitted in slides used as negative controls. Subsequently, slides were incubated with primary antibody amplifier (Thermo Fisher Scientific, Waltham, MA, USA) for 10 min, and with HRP Polymer Quanto (Thermo Fisher Scientific, Waltham, MA, USA) for an additional 10 min. After staining with 3,3′-diaminobenzidine (DAB) chromogen (Epredia, Portsmouth, NH, USA) for 3 min and counterstaining with Gill’s hematoxylin (DiaPath S.p.A, Martinengo, Bergamo, Italy) for 1 min, slides were rinsed in 0.02% ammonia solution for 30 s. Finally, slides were dehydrated using ethanol (96% and 100%) and xylol solutions, mounted with mounting medium (Richard-Allan Scientific, San Diego, CA, USA), and analyzed using light microscopy. Immunohistochemistry specificity was assessed using negative and positive controls of human liver, tonsil, and kidney for MMP-2, MMP-9, and VEGFA, respectively.

#### 2.2.2. YKL40 Immunohistochemistry

For YKL40 IHC evaluation, heat-induced antigen retrieval was performed using Epitope Retrieval Solution pH 9 (Leica Microsystems, Wetzlar, Germany), as previously described. Subsequently, endogenous peroxidase was blocked for 10 min, with 30% hydrogen peroxide in methanol. To decrease background signal due to endogenous avidin, biotin, or biotin-binding proteins, a sequential incubation with avidin block (BioLegend, San Diego, CA, USA) and biotin block (BioLegend, San Diego, CA, USA) (10 min each) was performed. Tissues were then incubated for 10 min with tissue blocker UltraVision Protein Block (Thermo Fisher Scientific, Waltham, MA, USA). Incubation with the YKL40 primary antibody [(AF2599, R&D Systems, Minneapolis, MN, USA); dilution 1:300] was carried out overnight at 4 °C. The primary antibody was omitted in the slides used as negative control. Tissue sections were further incubated for 10 min with a biotinylated secondary antibody, polyclonal rabbit anti-goat immunoglobulins (Dako, Glostrup, Denmark; dilution 1:200), and subsequently with ready-to-use (R.T.U.) horseradish peroxidase streptavidin (Vector, Newark, CA, USA) for 30 min. Human tonsil was used as positive control.

#### 2.2.3. Immunohistochemistry Evaluation and Scoring Method

IHC-stained tissue sections were analyzed for the presence and localization of the different proteins by a neuropathologist (blinded to the clinicopathological data of the patients and respective outcomes).

A semi-quantitative evaluation was carried out based on the following classifications from Table 1.

An immunohistochemistry score (IHS) was calculated for each antibody, using the previous information, and based on the following formula [33,34,35]:*IHS = “staining intensity” × “percentage of positive cells”*

### 2.3. Data and Statistical Analysis

For the statistical analysis of the IHC results, the IBM SPSS Statistics 28.0.0.0 software was used. Clinicopathological characteristics and IHC results are described as frequencies and respective percentages, or as medians and ranges. The assessment of significant associations between the IHC-evaluated proteins and patients’ clinicopathological features and survival data was assessed using McNemar’s test, Pearson’s chi-square, Fisher’s exact test and the Mann–Whitney U test. Survival distributions were estimated using Kaplan–Meier curves and statistical significances between groups were evaluated using the log-rank test. Multivariate survival analysis was conducted using Cox regression to establish independent prognostic factors. *p* < 0.05 was considered statistically significant.

## 3. Results

### 3.1. Evaluation of the Expression of Secretome-Associated Proteins in GB Patient Samples

#### 3.1.1. Clinicopathological Features of GB Patients

We performed a retrospective analysis of 63 glioblastoma patients from the Centro Hospitalar Universitário de São João (CHSUJ), diagnosed from 2011 to 2017 (clinical features are summarized in Table 2).

The median age at diagnosis was 56 years (26–73) and the disease was similarly prevalent in female and male patients (46% and 54%, respectively). Most of the patients presented a unilateral and unifocal lesion, mainly located in the temporal region of the brain. At diagnosis, most patients did not exhibit any severe functional impairments or difficulties with self-care, as evaluated according to the Eastern Cooperative Oncology Group (ECOG) performance scale and the Karnofsky Performance Scale (KPS). Regarding surgical procedures, total and partial/subtotal resections were the main interventions, with only four biopsies described. This group of patients received a median of six (0–42) cycles of temozolomide (TMZ) and a median of nine (1–41) doses of bevacizumab (10 mg/kg). All patients experienced recurrence at least twice during treatment, and their median overall survival was 23 months.

#### 3.1.2. IHC Analysis of MMP-2, MMP-9, VEGFA, and YKL40 Expression in GB Patients’ Tumor Tissues

The expression (and localization) of MMP-2, MMP-9, VEGFA, and YKL40 was evaluated in GB tissue samples by IHC, according to the classification presented in Table 1 (in the Materials and Methods Section).

The expression of MMP-2 was mainly found in neoplastic cells (mostly in cytoplasm), as observed in Figure 1a,b. MMP-2 expression was also found to a lesser extent in macrophages, necrotic tissue, microvascular proliferation (MVP), and/or endothelial cells.

The semiquantitative evaluation of MMP-2 expression (Table 3) revealed that 69.5% (41 out of 59) of the samples had a high percentage of positive cells, and 55.9% (33 out of 59) had an intense MMP-2 staining pattern. Additionally, 52.5% (31 out of 59) of the samples had the maximum IHC score value (HIS = 12).

The evaluation of MMP-9 showed that of the 61 samples evaluated, 52 (85.2%) were negative for MMP-9 in neoplastic cells (Table 3). Only 9 of 61 cases showed positive (cytoplasmic) staining, ranging from weak (3.3%) to strong (3.3%) intensity. However, none of these positive cases had more than 50% positivity for MMP-9 (Figure 1c,d). MMP-9 expression was mainly found in inflammatory cells, MVP, and necrotic tissue.

VEGFA expression was mainly found in the cytoplasm of neoplastic cells (Figure 1e,f). A total of 31 of 59 samples had over 50% of positively stained tumor cells, although only five samples (8.5%) showed strong intensity (Table 4). Mostly, the evaluated tissues had moderate or weak staining (39.0% and 35.6%, respectively). Regarding the IHS, our samples mostly had a low score (≤2) or an intermediate score (six or eight).

In our evaluation of YKL40, 28 out of 55 (50.9%) samples had a low percentage of positive YKL40 tumor cells (<25%), while 12 out of 55 (21.8%) had a high percentage (>75%) (Table 4). The majority of YKL40 expression (70.9%) was observed to be strong (and cytoplasmatic) (Figure 1g,h), as has been previously reported [36,37,38]. Moreover, YKL40 expression was observed, albeit to a lesser extent, in inflammatory cells.

#### 3.1.3. Association of Protein Expression in GB Tissues with Patients’ Clinicopathologic Features and Survival Data

To further evaluate the impact of the expression of MMP-2, MMP-9, VEGFA, and YKL40, an IHS threshold value for each protein was chosen based on the median value of their IHC score (as shown in Table 5).

We evaluated the relationship between the proteins studied that are known to be involved in similar biological processes or to regulate each other’s expression. To the best of our knowledge, no previous study has reported an association between all four proteins (MMP-2, MMP-9, VEGFA, and YKL40) in the context of GB, although some studies have shown associations for some of these proteins [39,40,41,42,43,44,45,46]. To further evaluate these potential associations in our series, McNemar’s test was performed using the threshold values from Table 5 (as shown in Appendix A). The significant associations listed in Appendix A all include MMP-9, likely due to the low number of cases with MMP-9 positivity in our series.

We then assessed the correlation between the expression of MMP-2, MMP-9, VEGFA, and YKL40 and the patients’ clinicopathological data, using Pearson’s chi-square or Fisher’s exact test and the Mann–Whitney U test (Table 6 and Appendix A).

The expression of all studied proteins was independent of patients’ sex and age at diagnosis (Appendix A). Moreover, our findings indicated no association between the levels of the evaluated proteins and patients’ performance status, tumor laterality, focality, type of surgery, or MGMT methylation status (Appendix A). Only the expression levels of YKL40 showed an association with the anatomical location of the tumor in the brain (*p* = 0.007, Table 6). To the best of our knowledge, this association has not been reported in any study so far.

Furthermore, our results showed an association between higher expression of YKL40 and an increased median number of bevacizumab doses administered (*p* = 0.035, Table 6), which rose from 6 in patients with low YKL40 tumor expression in their primary tumor to 9.5 in patients with high expression.

#### 3.1.4. Impact of MMP-2, MMP-9, VEGFA, and YKL40 Expression in GB Patients’ Survival

The impact of the studied proteins on GB patients’ progression-free survival (PFS) and overall survival (OS) was evaluated using Kaplan–Meier analyses. To do so, two scenarios were taken into consideration regarding PFS analysis: (i) the first, the progression-free survival after TMZ treatment until the first recurrence (PFS-1); and (ii) the second, the progression-free survival after bevacizumab treatment until the second recurrence (PFS-2).

Although we have evaluated the effect of all the studied proteins on patients’ outcomes (Appendix A), focus will be given to the most relevant results, which in this instance are from VEGFA (target for bevacizumab therapy) and also YKL40 (associated with the number of bevacizumab doses in Table 6). Additionally, a previous association between VEGFA and YKL40 has been established in the context of antiangiogenic treatment for GB [39].

VEFGA expression had no significant impact on PFS-1 (Figure 2a), although a tendency for the concomitant expression of MMP-2 and VEGFA was observed (*p* = 0.061, Appendix A), with PFS-1 increasing from 8 (5–20) months to 12 (5–32) months for low concomitant expression of MMP-2 and VEGFA.

When addressing the impact of VEGFA in PFS-2 (after treatment with bevacizumab), VEGFA was the only protein whose expression levels (individually) had a significant impact on patients’ outcomes (*p* = 0.029, Figure 2b). VEGFA high expression significantly associated with increased PFS-2 (10 months) compared with VEGFA low expression (4 months). Figure 2b represents the results obtained based on the IHS threshold, but the same findings were obtained when assessing the impact of VEGFA staining intensity and percentage of positive cells, individually (Appendix A). Patients with tumor tissues presenting moderate/strong VEGFA staining intensity, or over 50% VEGFA-positive cells, showed a longer period without recurrence after bevacizumab treatment (PFS-2).

Moreover, there was a clear impact of VEGFA concomitant expression with other studied proteins in PFS-2. This was particularly relevant when VEGFA was combined with YKL40 (Figure 3a), with high concomitant expression of VEGFA/YKL40 significantly associating (*p* = 0.009) with longer survival of patients after bevacizumab treatment (PFS-2). The same was observed for MMP-9 (*p* = 0.039, Figure 3b), and to a lesser extent, for the concomitant expression of VEGFA and MMP-2 (*p* = 0.063, Figure 3c). The positive effect of the high concomitant expression of VEGFA with the remaining proteins on patients’ PFS-2 seems to be driven by the expression of VEGFA; this reinforces its potential as a predictor of progression-free survival after bevacizumab treatment.

Regarding YKL40 expression, no statistically significant results were obtained for PFS-1 (*p* = 0.481, Figure 4a) or for PFS-2 (*p* = 0.183, Figure 4b). However, there was a tendency for the high expression of YKL40 to associate with longer survival of patients after bevacizumab treatment (PFS-2; Figure 4b).

Additionally, we performed an analysis of the statistical differences concerning the median expression of VEGFA and YKL40 between good responders (PFS-2 > 6 months) and bad responders (PFS-2 ≤ 6 months) to bevacizumab treatment (Table 7), according to the previously reported survival threshold values [47,48].

This analysis revealed that good responders to bevacizumab presented a higher median expression of VEGFA compared to bad responders (*p* = 0.035). Concerning YKL40 expression, the analysis also showed a tendency towards a higher YKL40 expression among good responders, despite the non-significancy (*p* = 0.069). 

Based on the results showing VEGFA as a predictor biomarker for bevacizumab, we carried out a multivariate analysis of covariates on PFS-2 (including age, ECOG, type of surgery and VEGFA expression). VEGFA expression revealed a tendency (*p* = 0.075) but without statistical significance for independent predictors of patients’ progression-free survival after bevacizumab (Table 8).

Regarding overall survival (OS), no clear impact was observed based on the individual expression of the studied proteins (Appendix A). Our results showed only the concomitant expression of MMP-2 and MMP-9 to significantly impact patients’ survival (*p* = 0.046, Appendix A). 

## 4. Discussion

Despite the increasing knowledge of its underlying mechanisms, GB remains a dreadful disease with patients presenting limited survival. Moreover, the lack of effective therapeutic strategies reinforces the need to identify and validate biomarkers that can provide clinically useful insights regarding the course of the disease and/or patients’ outcome.

GB secretome holds tremendous potential as a promising source of GB biomarkers. Among several GB secretome molecules, a group of four proteins (MMP-2, MMP-9, VEGFA, and YKL40) emerged in the literature, based on their role in GB aggressiveness and because they have been found differentially expressed in the plasma of GB patients [20,30,31,49,50,51,52]. This could bring new opportunities for their evaluation not only in tumor tissues but also in patients’ plasma, although we did not address this possibility in the present study.

We evaluated the prognostic value of these four proteins in patients’ tumor samples of a retrospective cohort of GBs, treated with TMZ (Stupp protocol) in first line, and with bevacizumab upon recurrence (second line).

Our study revealed promising results regarding the potential of VEGFA as a prognostic biomarker for the response to bevacizumab treatment (PFS-2). As previously reported in some studies [43,53,54,55,56], VEGFA was mostly found in neoplastic cells in our series of samples. Despite not being evident in our results, VEGFA is also known to be present in endothelial cells, being stimulated by hypoxia (via HIF pathway) [15]; its expression has been reported in blood vessels as well [53,57,58], and near foci of necrosis (hypoxic areas) [59].

A higher tumor VEGFA expression may indicate a better response to bevacizumab, and when combined with the expression of either one of the other proteins (particularly with YKL40), VEGFA expression is even more promising. There is a limited number of studies addressing specifically VEGFA and PFS-2 in GB patients, particularly in our setting, and there are some inconsistencies in the conclusion of clinical trials (carried out in patients treated with bevacizumab). One trial showed that anti-VEGFA treatment did not impact patients’ survival, while another demonstrated an increased PFS associated with the bevacizumab-treated cohort [9,60]. Moreover, García-Romero et al. reported that patients with increased baseline levels of VEGFA in their serum were more responsive to bevacizumab therapy, since more VEGFA is available to be targeted by bevacizumab [61].

Although VEGFA is the target of bevacizumab [62], and its impact in patients’ PFS-2 could be expected, there is still limited information regarding the effect of VEGFA expression specifically in cohorts such as the one used in our study (i.e., patients treated with bevacizumab after recurrence upon TMZ treatment). A study evaluating tumor tissues from patients treated with bevacizumab plus irinotecan showed that high VEGFA expression was associated with higher probability of response to radiotherapy [63]. Results from a phase III trial evaluating the effect of bevacizumab in combination with standard chemoradiotherapy (and maintenance with adjuvant TMZ) did not show a significant improvement in patients’ survival [9], while another phase III trial (in an identical clinical setting) revealed a higher PFS for the bevacizumab-treated group compared to the control group, although no differences were observed in patients’ OS [60]. 

Additionally, high VEGFA expression was associated with good responders to bevacizumab treatment (PFS-2 > 6 months), and although the multivariate analysis did not yield significant results, VEGFA expression tended to be an independent predictor of PFS-2 in GB patients.

Our results showed YKL40 to be highly expressed in tumor cells, consistent with previous reports [36,37,38]. Inflammatory cells also showed some expression of YKL40, which has been previously attributed to the presence of tumor-infiltrating macrophages [22,64]. A role of YKL40 in the MGMT-associated resistance to alkylating agents has been suggested [65]; furthermore, a change in the (gene) functions of YKL40 in GBs according to the methylation status of MGMT has been reported [22]. However, this association between YKL40 and MGMT was not reflected in our series of GBs. While high levels of YKL40 did not significantly impact patients’ prognoses, they were found to significantly associate with a greater number of bevacizumab doses administered upon recurrence. This suggests that patients with high YKL40 tumor expression may experience a longer period without recurrence, possibly due to an increased response/sensitivity to bevacizumab (with patients undergoing an extended treatment). Previous studies have evaluated the effect of YKL40 levels after antiangiogenic therapy but did not specifically focus on the number of bevacizumab doses administered. Although some of these studies are not in agreement with our results, it is important to consider what is being evaluated—particularly the type of samples analyzed—as well as the characteristics of the patients involved (including the therapeutic regimen). In 2018, Boisen and colleagues reported an improvement in patients’ outcome—particularly for progression-free survival (PFS)—after bevacizumab treatment for patients with low baseline YKL40 plasma levels [49]; similar results were observed in ovarian cancer patients [66]. However, long periods of bevacizumab treatment have been shown to upregulate YKL40 in GB mice xenograft models [39], which suggests a significant role of YKL40 in the response to bevacizumab, particularly in terms of resistance to the antiangiogenic therapy.

In our series of GBs, MMP-2 and MMP-9 expression had little impact on patients’ outcome. MMP-2 was found to be highly expressed in almost all tissue samples, not only in tumor cells (mostly), but also in inflammatory cells, endothelial cells, MVP, and necrotic areas. This is consistent with previous studies, which have described a high expression of MMP-2 in GB tissues [23,24,35,67,68], and tumor cells as its main source [24,35]. MMP-9 was mostly absent in tumor cells and mostly present in inflammatory cells, necrotic areas, and MVP. MMP-9 expression in GB tissues is still a matter of debate, with some studies describing its absence in neoplastic cells [31,69] and others showing its increase [35,45,68,70]. High MMP-9 expression among inflammatory cells has been previously described in other studies [71,72,73], being most likely associated with neuroinflammatory processes [73] and/or secretion of its inactive form by part of the inflammatory cell populations [74]. Jiguet-Jiglaire and colleagues recently demonstrated that MMP-9 is expressed by tumor-infiltrating neutrophils in GB tissue [31]. Regarding the impact on patients’ survival, MMP-2 and MMP-9 were only relevant for OS when concomitantly expressed, which might be expected considering their described association [35,68].

It is important to note that this is a retrospective study with a limited number of patient samples, for which not all the clinicopathological information was available, which may have hindered some associations. Another constraint, related to the small number of cases we evaluated, was that we were limited to analyzing mostly primary tumor samples and unable to analyze tumor recurrence samples. Since tumor recurrence samples could provide valuable insights into tumor heterogeneity during different phases of tumor progression, it would be important to evaluate these in future studies.

## 5. Conclusions

In conclusion, the findings of our study suggest that VEGFA has the potential to be used as a tissue biomarker for predicting patients’ response to bevacizumab, and could therefore be useful for personalized treatment. The results also demonstrated that YKL40 may have a role in the response to bevacizumab in GB therapy. Further validation of these findings with larger patient cohorts is necessary. Additionally, exploring the levels of these proteins in circulating fluids such as plasma could provide valuable information regarding their potential as circulating biomarkers.

## Figures and Tables

**Figure 1 cancers-15-02196-f001:**
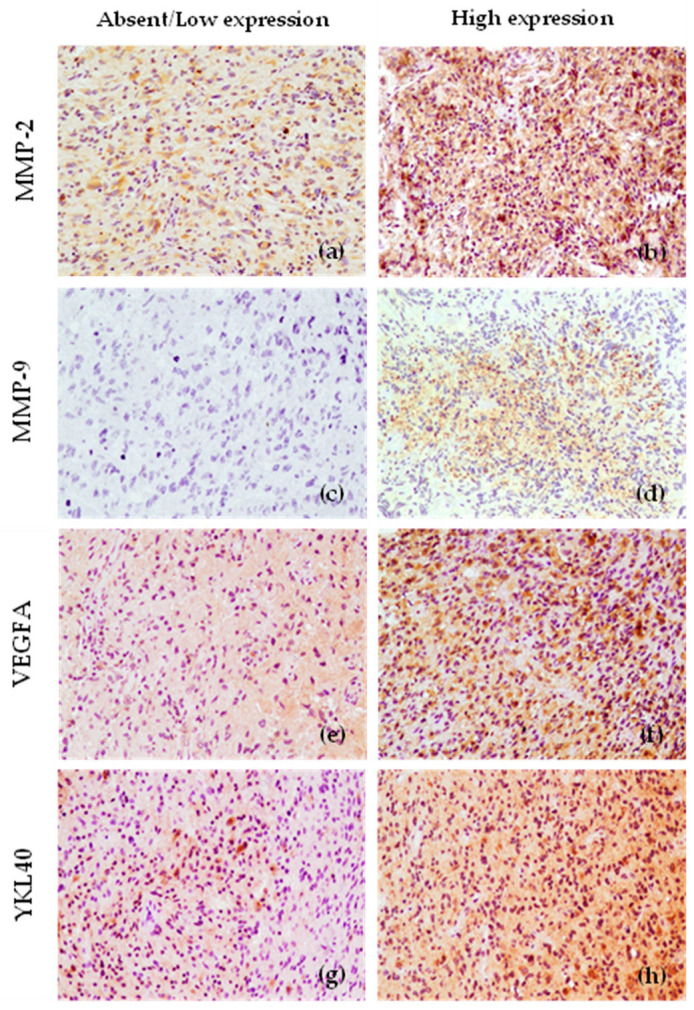
Representative images of the IHC analysis of MMP-2 (**a**,**b**), MMP-9 (**c**,**d**), VEGFA (**e**,**f**), and YKL40 (**g**,**h**) in human GB samples with absent/low (**a**,**c**,**e**,**g**) and high expression (**b**,**d**,**f**,**h**) of each protein. Magnification: 200×.

**Figure 2 cancers-15-02196-f002:**
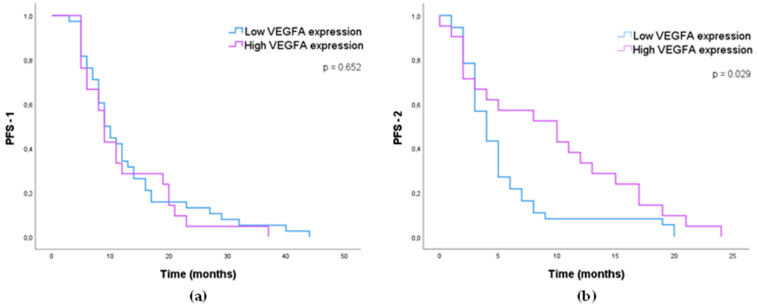
Kaplan–Meier analysis of progression-free survival (in months) after temozolomide treatment (PFS-1) (**a**) and bevacizumab treatment (PFS-2) (**b**) according to VEGFA expression.

**Figure 3 cancers-15-02196-f003:**
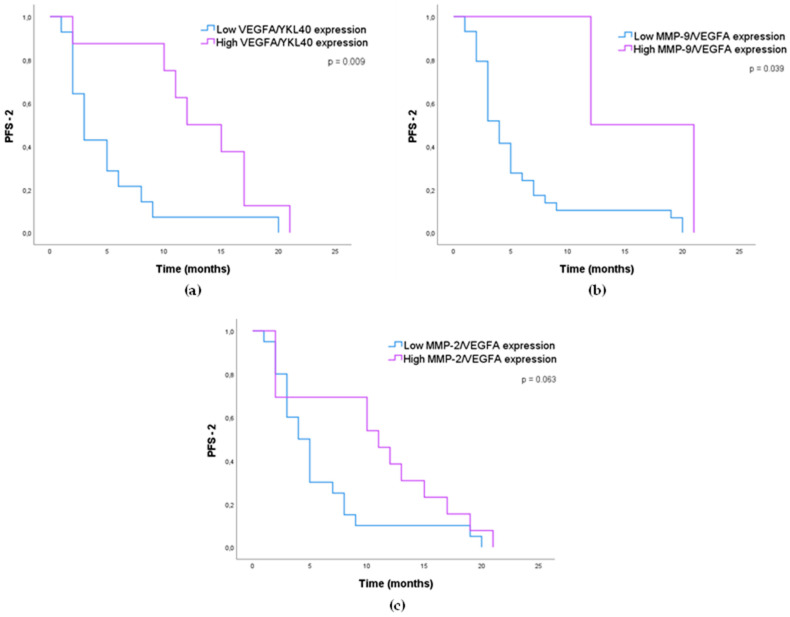
Kaplan–Meier analysis of progression-free survival (in months) after bevacizumab treatment (PFS-2) according to the concomitant expression of VEGFA with YKL40 (**a**), MMP-9 (**b**), and MMP-2 (**c**).

**Figure 4 cancers-15-02196-f004:**
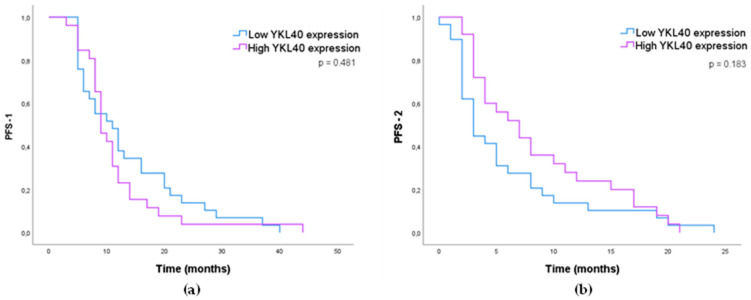
Kaplan–Meier analysis of progression-free survival (in months) after temozolomide treatment (PFS-1) (**a**) and bevacizumab treatment (PFS-2) (**b**) according to the expression of YKL40.

**Table 1 cancers-15-02196-t001:** Classification of the intensity of the IHC staining (**a**) and the proportion of positive cells (**b**).

(a)		(b)		
Staining Intensity		Percentage of Positive Cells		
		MMP-2 and MMP-9		VEGFA and YKL40
Absent Weak Moderate Strong	0 1 2 3	0% 1−5% 6−50% 51−75% >75%	0 1 2 3 4	0% 1−24% 25−49% 50−75% >75%

**Table 2 cancers-15-02196-t002:** Summary of GB patients’ clinical data/features.

Clinical Features	n (%)
**Sex** [n = 63]	
Male	34 (54.0%)
Female	29 (46.0%)
**Age at diagnosis** (years) [n = 63]	
Median (range)	56 (26–73)
**Type of surgery** [n = 57]	
Biopsy	4 (7.0%)
Partial or subtotal resection	26 (45.6%)
Total resection	27 (47.4%)
**Tumor location** [n = 63]	
Frontal	12 (19.0%)
Temporal	21 (33.3%)
Parietal	9 (14.3%)
Multiple	17 (27.0%)
Other	4 (6.3%)
**Tumor laterality** [n = 62]	
Unilateral	60 (96.8%)
Bilateral	2 (3.2%)
**Tumor focality** [n = 60]	
Unifocal	55 (91.7%)
Multifocal or multicentric	5 (8.3%)
**ECOG** [n = 61]	
0	13 (21.3%)
1	42 (68.9%)
2	5 (8.2%)
3	1 (1.6%)
**KPS** [n = 61]	
<70	1 (1.6%)
≥70	60 (98.4%)
**MGMT Status** [n = 40]	
Methylated	20 (50.0%)
Non-Methylated	20 (50.0%)
**Number of temozolomide cycles** [n = 63]	
Median (range)	6 (0–42)
**Number of bevacizumab doses** [n = 63]	
Median (range)	9 (1–41)
**Progression-free survival after temozolomide** (months) [n = 63]	
Median (range)	9 (3–44)
**Progression-free survival after bevacizumab** (months) [n = 62]	
Median (range)	4 (0–24)
**Overall survival** (months) [n = 63]	
Median (range)	23 (1–63)
**Patient status** [n = 63]	
Dead	52 (82.5%)
Alive or Unknown	11 (17.5%)

The study was conducted on 63 patients. The varying *n* values in front of each feature are due to incomplete clinical data in some cases, as indicated. *ECOG*—Eastern Cooperative Oncology Group Performance Status Scale; *KPS*—Karnofsky Performance Scale.

**Table 3 cancers-15-02196-t003:** Percentage of positively stained cells, staining intensity, and staining score (IHS) of MMPs-2 and -9 evaluated by IHC in GB tumor tissues.

	MMP-2 (*n* = 59)	MMP-9 (*n* = 61)
**Percentage of positive stained cells**		
0%	0 (0.0%)	52 (85.2%)
1–5%	1 (1.7%)	8 (13.1%)
6–50%	7 (11.9%)	1 (1.6%)
51–75%	10 (16.9%)	0 (0.0%)
>75%	41 (69.5%)	0 (0.0%)
**Staining intensity**		
Absent	0 (0.0%)	52 (85.2%)
Weak	3 (5.1%)	2 (3.3%)
Moderate	23 (39.0%)	5 (8.2%)
Strong	33 (55.9%)	2 (3.3%)
**IHS**		
0	0 (0.0%)	52 (85.2%)
1	1 (1.7%)	2 (3.3%)
2	1 (1.7%)	4 (6.6%)
3	1 (1.7%)	2 (3.3%)
4	5 (8.5%)	1 (1.6%)
6	9 (15.3%)	0 (0.0%)
8	10 (16.9%)	0 (0.0%)
9	1 (1.7%)	0 (0.0%)
12	31 (52.5%)	0 (0.0%)

**Table 4 cancers-15-02196-t004:** Percentage of positively stained cells, staining intensity, and staining score (IHS) of VEGFA and YKL40, evaluated by IHC in GB tumor tissues.

	VEGFA (*n* = 59)	YKL40 (*n* = 55)
**Percentage of positive stained cells**		
0%	10 (16.9%)	8 (14.5%)
1–24%	10 (16.9%)	20 (36.4%)
25–49%	8 (13.6%)	6 (10.9%)
50–75%	15 (25.4%)	9 (16.4%)
>75%	16 (27.1%)	12 (21.8%)
**Staining intensity**		
Absent	10 (16.9%)	8 (14.5%)
Weak	21 (35.6%)	1 (1.8%)
Moderate	23 (39.0%)	7 (12.7%)
Strong	5 (8.5%)	39 (70.9%)
**IHS**		
0	10 (16.9%)	8 (14.5%)
1	8 (13.6%)	1 (1.8%)
2	8 (13.6%)	5 (9.1%)
3	4 (6.8%)	14 (25.5%)
4	4 (6.8%)	1 (1.8%)
6	10 (16.9%)	6 (10.9%)
8	11 (18.6%)	0 (0.0%)
9	2 (3.4%)	8 (14.5%)
12	2 (3.4%)	12 (21.8%)

**Table 5 cancers-15-02196-t005:** Distribution of cases according to the IHS threshold value chosen for each protein.

Protein	IHS Threshold	
**MMP-2**	**<12**	**12**
	28 (47.5%)	31 (52.5%)
**MMP-9**	**0**	**≥** **1**
	52 (85.2%)	9 (14.8%)
**VEGFA**	**≤** **4**	**>4**
	38 (64.4%)	21 (35.6%)
**YKL40**	**≤** **4**	**>4**
	29 (52.7%)	26 (47.3%)

**Table 6 cancers-15-02196-t006:** Association of clinicopathological features of GB patients with the expression levels of YKL40.

Clinical Features	YKL40		
	IHS ≤ 4	IHS > 4	*p* Value
**Tumor Location** Frontal Temporal Parietal Multiple Others	***n* = 57**		**0.007**
	5 (17.2%) 8 (27.6%) 1(3.4%) 12 (41.4%) 3 (10.3%)	5 (19.2%) 10 (38.5%) 8 (30.8%) 3 (11.5%) 0 (0.0%)	
**Number of Bevacizumab Doses** Median (range)	***n* = 55**		**0.035**
	6 (1–37)	9.5 (2–41)	

Only results which were significant for the correlation between YKL40 protein expression levels (based on the IHS threshold value) and clinicopathological features of GB patients are presented. Similar analyses were carried out for MMP-2, MMP-9, and VEGFA. The complete set of results is presented in Appendix A.

**Table 7 cancers-15-02196-t007:** Mann–Whitney U test analysis of statistical differences regarding the expression of VEGFA and YKL40 between good responders (PFS-2 > 6 months) and bad responders (PFS-2 ≤ 6 months) to bevacizumab treatment.

	VEGFA Expression	YKL40 Expression
**Mann** **–Whitney U**	252.500	258.000
***p* value**	**0.035**	0.069

**Table 8 cancers-15-02196-t008:** Multivariate analysis of independent predictors of progression-free survival after bevacizumab treatment (PFS-2) of GB patients.

Variable	Multivariate Analysis		
	HR	95% CI	*p* Value
**Age** <65 years versus ≥65 years	1.37	0.76–2.45	0.294
**ECOG** 0–1 versus ≥ 2	1.16	0.48–2.83	0.745
**Type of surgery**	0.77	0.23–2.64	0.682
VEGFA expression	**0.57**	**0.31–1.06**	**0.075**

*HR Hazard ratio, CI Confidence interval.*

## Data Availability

Not applicable.

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
