# Peer review of "High VEGFA Expression Is Associated with Improved Progression-Free Survival after Bevacizumab Treatment in Recurrent Glioblastoma"

_cancers, 2023, doi:10.3390/cancers15082196_

Round 1
Reviewer 1 Report
We dont take any consideration to the pseudo-progression and we can observe a pseudo-progression frequently to the rGB (recurrent GB). If this apply then the PFS is not apply for pValue conclusions.
I agree that we need to focus to the rGB tissues and not from newly diagnosed GB for a large study to confirm the data.
Author Response
Dear Reviewer,
We thank you for taking the time to evaluate our manuscript and for your careful comments. We hope that we have adequately addressed them, as follows:
Comment 1) "We dont take any consideration to the pseudo-progression and we can observe a pseudo-progression frequently to the rGB (recurrent GB). If this apply then the PFS is not apply for pValue conclusions."
In our MDT meeting, two neuroradiologists analyze all MRIs and carefully evaluate both contrast-enhanced T1 and T2/FLAIR sequences, taking into consideration several aspects such as corpus callosum involvement, subependymal spread, multiple enhancing lesions, infiltration of the cortical ribbon or location outside of the radiation field. Additionally, perfusion MRI is routinely used to help differentiate tumour progression from pseudoprogression. If doubt persists, PET scan is also used. Considering this, only after thorough imaging investigation were tumor recurrences used to calculate PFS. In this perspective, PFS was used as a viable outcome as it is used in numerous studies and clinical trials.
Comment 2)"I agree that we need to focus to the rGB tissues and not from newly diagnosed GB for a large study to confirm the data."
We agree. The fact that it was not possible to analyse tumoural recurrence samples due to the small number of cases is a limitation. This important aspect, that reflects the tumour heterogeneity in the different phases of the disease, needs to be addressed in further studies. We are pleased to inform you that this aspect is already being included in ongoing research projects.
While we have addressed this issue in the initial manuscript, we have taken your feedback into consideration and made additional alterations to the text to ensure that it this is more clear. (please see line 442).
Reviewer 2 Report
The research article "High VEGFA expression is associated with improved progression free survival after bevacizumab treatment in recurrent glioblastoma" authored by Alves et al., is very good study. The experiments performed by the authors were very elaborate and successful in achieving the specific aims defined in the manuscript. The results were very clear and nicely presented. The study would interest a broad readership group.
Author Response
We thank you for taking the time to review our research article and for your positive comments.
We greatly appreciate your feedback and are pleased to hear that you found our study to be well-conducted and the results to be clear and nicely presented. We aimed to achieve specific aims defined in the manuscript, and we are glad to know that you found that we were successful in doing so.
Reviewer 3 Report
Authors evaluated the relationship between MMP-2, MMP-9, VEGFA, and YKL40 expression in glioblastoma tissues from patients treated with temozolomide or bevacizumab. This is a retrospective study which was conducted on a limited number of patient samples, whose clinicopathological data are not all available. Despite this, the study is well conducted and nonetheless, for the first time the expression of these four proteins was simultaneously measured in glioblastoma.
Author Response
Thank you for taking the time to review our study and for your positive feedback. We are pleased to know that despite the limitations that we also refer in the manuscript, you found our study to be well-conducted and our results valuable.